# Differential circulating miRNA profiles identified miR-423-5p, miR-93-5p, and miR-4532 as potential biomarkers for cholangiocarcinoma diagnosis

Kittiya Supradit[1,*], Sattrachai Prasopdee[2,3,4,*], Teva Phanaksri[2], Sithichoke Tangphatsornruang[5], Montinee Pholhelm[2,3,4], Siraphatsorn Yusuk[4], Kritiya Butthongkomvong[6], Kanokpan Wongprasert[7], Jutharat Kulsantiwong[8], Amnat Chukan[9], Smarn Tesana[3] and Veerachai Thitapakorn[2,3,4]

[1] Radiological technology, Faculty of Science, Ramkhamhaeng University, Bangkok, Thailand
[2] Chulabhorn International College of Medicine (CICM), Thammasat University, Pathum Thani, Thailand
[3] Research Group in Multidimensional Health and Disease (MHD), Chulabhorn International College of Medicine, Thammasat University, Pathum Thani, Thailand
[4] Thammasat Research Unit in Opisthorchiasis, Cholangiocarcinoma, and Neglected parasitic Diseases (TRU-OCN), Thammasat University, Pathum Thani, Thailand
[5] National Center for Genetic Engineering and Biotechnology, National Science and Technology Development Agency, Pathum Thani, Thailand
[6] Medical Oncology, Udon Thani Cancer Hospital, Udon Thani, Thailand
[7] Department of Anatomy, Faculty of Science, Mahidol University, Bangkok, Thailand
[8] Faculty of Science, Udon Thani Rajabhat University, Udon Thani, Thailand
[9] Prima Scientific Co. Ltd, Bangkok, Thailand
* These authors contributed equally to this work.

Corresponding author
Veerachai Thitapakorn, veebkk@gmail.com

## ABSTRACT

**Background:** Cholangiocarcinoma (CCA) is high in morbidity and mortality rates which may be due to asymptomatic and effective diagnostic methods not available. Therefore, an effective diagnosis is urgently needed.

**Methods:** Investigation of plasma circulating miRNA (cir-miRNA) was divided into two phases, including the discovery phase (pooled 10 samples each from three pools in each group) and the validation phase (17, 16, and 35 subjects of healthy control (HC), *O. viverrini* (OV), and CCA groups, respectively). The plasma from healthy control subjects, *O. viverrini* infected subjects, and CCA subjects was used. In the discovery phase, plasma was pooled by adding an equal volume of plasma, and cir-miRNA was isolated and analyzed with the nCounter® SPRINT Profiler. The significantly different cir-miRNAs were selected for the validation phase. In the validation phase, cir-miRNA was isolated and analyzed using real time-quantitative polymerase chain reaction (RT-qPCR). Subsequently, statistical analysis was conducted, and diagnostic parameters were calculated.

**Results:** Differential plasma cir-miRNA profile showed at least three candidates including miR-423-5p, miR-93-5p, and miR-4532 as potential biomarkers. From validation of these cir-miRNAs by RT-qPCR, the result showed that the satisfied sensitivity and specificity to differential CCA group from HC and OV group was obtained from miR-4532 ($P < 0.05$) while miR-423-5p and miR-93-5p can be used for differential CCA from OV and HC group ($P < 0.05$) with high specificity but limited

the sensitivity. In conclusion, candidate cir-miRNAs have been identified as potential biomarkers including miR-423-5p, miR-93-5p and miR-4532. Screening by miR-4532 and confirmed with miR-423-5p, miR-93-5p were suggested for differential CCA patients in the endemic area of *O. viverrini*.

# INTRODUCTION

Cholangiocarcinoma (CCA), the cancer of the bile duct, is associated with high morbidity and mortality (*Halder, Amaraneni & Shroff, 2022*). One of the main risk factors for CCA is liver fluke infection, including *Opisthorchis viverrini*, *Opisthorchis felineus*, and *Clonorchis sinensis*, which have been identified as carcinogens by IACR (*IARC, 2012*). Chronic infection by these liver flukes is mostly asymptomatic, allowing the development of a severe form, cholangiocarcinoma (*Brindley et al., 2021*).

The age-standardized incidence rate of CCA has been reported to be 85, 8.8, and 7.6 cases per 100,000 in Northeast-Thailand, Gwangju-Korea, and Shanghai-China, respectively. In contrast, the rest of the world has an incidence rate of less than 4.7 cases per 100,000 (*Khan, Tavolari & Brandi, 2019*; *Brindley et al., 2021*; *Banales et al., 2016*). Various risk factors are associated with CCA, including infections, socio-cultural practices, diet and lifestyle, and environmental factors (*Songserm et al., 2021*). Several infections have been identified as risk factors for CCA, such as *O. viverrini*, *C. sinensis*, *O. felineus*, and hepatitis B and C viruses. Furthermore, the combination of other risk factors such as nitrosamine compounds, pesticides, and vulnerable sources of drinking water also contribute to the risk (*Pupacdi et al., 2023*). The combination of liver fluke infection in endemic areas such as Thailand and nitrosamine compounds in pickled or fermented foods are responsible for the increased risk of CCA. Chronic and repeated liver fluke infections stimulate inflammation and oxidative stress, while nitrosamine compounds induce the alkylation of DNA, both of which can cause DNA damage (*Sripa, Tangkawattana & Brindley, 2018*; *Pinlaor et al., 2004a*, *2004b*).

Most CCA patients visit the hospital in the late stage, limiting the efficacy of therapy (*Wang et al., 2021a*). This is due to CCA being mostly asymptomatic, with no effective diagnosis similar to *O. viverrini* infection (*Khuntikeo et al., 2018*). Several attempts have been done to develop an effective diagnostic test for both OV and CCA. Recently, liquid biopsy-based diagnosis, such as cell-free DNA, circulating miRNA (cir-miRNA), circulating tumor DNA (ctDNA), exosomal miRNA, and circulating tumor cells (CTCs), has become more attractive due to its minimally invasive approach (*Drula et al., 2020*). The miRNA is a small non-coding miRNA about 20–25 nucleotides in length which plays a key role in gene regulation by degrading the complementary mRNA. The miRNA is involved in various biological processes, including development, differentiation, and diseases including cancers. The miRNA was applied to several cancers for various approaches such as diagnosis, prognosis, and therapeutics. The specific miRNA species was used to
diagnose several cancers including CCA. However, application of miRNA for CCA is still limited, especially for *O. viverrini* related CCA.

From previous and our preliminary studies, the increase of plasma cell-free miRNA showed higher sensitivity and specificity compared to the increase of plasma cfDNA (*Prasopdee et al., 2024*) and ctDNA. Moreover, the detection of CTCs and exosomal miRNA is technically complicated, requiring expensive instruments and considerable time, making it unsuitable for routine diagnostic use in the future. Therefore, cir-miRNA is a potential candidate for further validation as a diagnostic biomarker. Therefore, the aim of this study is to investigate the biomarker potential of specific up-regulating miRNA for the development of a diagnostic tool for CCA. The Counter® SPRINT Profiler (NanoString Technologies, Seattle, WA, USA) was selected for differential profiling and identified the potential miRNA biomarkers, and real time-quantitative polymerase chain reaction (RT-qPCR) was then used for validation.

## MATERIALS AND METHODS

### Chemicals and reagents

EDTA blood tubes were purchased from Becton Dickinson. The miRNAeasy® Serum/Plasma Kit (Qiagen, Gaithersburg, MD, USA) was used for the extraction of plasma cir-miRNA. The measurement of miRNA concentration was done by using a microvolume spectrophotometer (NanoDrop 2000; Thermo Fisher Scientific, Greenville, DE, USA) and Qubit™ miRNA Assay Kit (Invitrogen, Waltham, MA, USA). The miRNA profile was analyzed by nCounter® SPRINT Profiler using Human V3 miRNA panel (NanoString Technologies, Seattle, WA, USA). The quantitation of miR-423-5p, miR-93-5p, and miR-4532 were investigated by ID3EAL miRNA qPCR Assay (MiRXES Pte Ltd, Singapore).

### Study cohort

This study cohort was collected during year 2017–2019 and approved by the Human Ethic Committee of Udonthani Cancer Hospital, Udon Thani, Ministry of Public Health, Thailand (UCH-CT 11/2563). All subjects received written informed consent and gave their approval before the study.

Since chronic *O. viverrini* (OV) infection is the etiology of CCA, detection of OV infection and treatment with praziquantel are essential to prevent CCA development. However, detecting OV infection has been challenging due to the limitations of the gold standard fecal examination sensitivity. According to a previous study, most patients infected with OV were asymptomatic (*Prasopdee et al., 2023*). When OV patients are not treated, chronic liver fluke infection and CCA can develop. Therefore, differential diagnosis of OV is a crucial step in preventing the development of CCA. In this study, therefore, subjects with OV infection and CCA were recruited to identify potential differential diagnostic biomarkers for OV/CCA as well as diagnostic biomarkers for CCA.

Briefly, recruitment process of the subjects was done based on health statuses including healthy control (HC), *O. viverrini* infected subject, and cholangiocarcinoma subject. The healthy control group, which included the subjects with a normal physical examination, no
liver enlargement, no jaundice, and who tested negative for *O. viverrini* egg on fecal examination; the group with *O. viverrini*-infected subjects, which included the subjects who tested positive for *O. viverrini* egg on fecal examination with normal liver and bile duct by ultrasonography; and the group with cholangiocarcinoma subjects, which included subjects with hepatomegaly and/or jaundice on physical examination, abnormal liver and bile duct ultrasonography, and confirmed cholangiocarcinoma by tissue histopathology. All the subjects in the OV group were asymptomatic. This study was divided into two phases including discovery phase by nCounter® SPRINT Profiler and validate phase by RT-qPCR.

Heterogeneity is commonly observed in various cancers. Individual samples can exhibit variability due to genetic, environmental, and lifestyle differences among subjects. To address this heterogeneity, plasma samples were consolidated by pooling to minimize variability, thereby providing a more representative sample for this study (*Schisterman et al., 2010*; *Kendziorski et al., 2005*). For the discovery phase, 10 µl of each of the 10 plasma samples of each group were pooled individually for three pools (30 plasma samples/group and 90 plasma samples in total) and further subjected for nCounter® SPRINT Profiler. For the validated phase, another 17 subjects of HC, 16 subjects of OV, and 35 subjects of CCA were recruited for RT-qPCR analysis. The demographics and clinical statuses of the subjects, including sex, age, alcohol consumption, smoking, eating habits of raw fish and fermented foods, and history of OV infection, were summarized in Table 1 for both the discovery and validation phases.

## Extraction of circulating miRNA

The cir-miRNA was isolated from 200 µl of plasma samples using miRNeasy Serum/Plasma Kit according to the manufacturer's protocol. Briefly, 1 ml QIAzol lysis reagent was added to 200 µl thawed plasma, mixed and incubated at room temperature for 5 min. A total of 3.5 µl Spike in control (MIRXES, Singapore) was added into the sample lysis buffer. A total of 200 µl chloroform was added, followed by shaking, incubation and centrifugation. The upper aqueous phase was transferred and 900 µl ethanol was added into a new tube and then transferred to the RNeasy MinElute column. The column was washed with RWT, RPE, and 80% Ethanol. following centrifugation to dry the membrane, RNAs were eluted with 14 µl RNase-free water. The isolated plasma circulating RNA (cfRNA) concentration was evaluated using the NanoDrop 2000 spectrophotometer.

## Investigation of cir-miRNA profiles

The 100 ng of isolated plasma RNA were processed following the manufacturer's manual. The cir-miRNA profile was analyzed by nCounter® SPRINT Profiler using Human V3 miRNA panel. The data of raw count was analyzed by the nCounter® SPRINT Profiler for image capture (190 fields of view). Analysis of cir-miRNA data was performed using nSolver Analysis (version 4.0) software. Subtraction of each cir-miRNA count data was done by using the geometric mean of the negative controls. For data integrity, the counting copy number of cir-miRNA with less than 25 were excluded. Profiling data were

**Table 1 Demography of subject.** The demographic and clinical statuses of participants.

| | Discovery phase by NanoString technology | | | Validation phase by qRT-PCR | | |
|---|---|---|---|---|---|---|
| Group (Sample size) | N | OV | CCA | N | OV | CCA |
| Sample No. | 30 | 30 | 30 | 17 | 16 | 35 |
| Sex (Male/Female) | | | | | | |
| Male | 4 | 16 | 22 | 2 | 13 | 20 |
| Female | 26 | 14 | 8 | 15 | 3 | 15 |
| Age (year) | | | | | | |
| Min/Max | 20/56 | 42/71 | 48/75 | 25/60 | 18/79 | 48/87 |
| Mean ± SD | 39.3 ± 9.8 | 56.4 ± 8.2 | 59.9 ± 6.8 | 37.2 ± 8.9 | 51.6 ± 16.6 | 65.2 ± 11.4 |
| Alcohol consumption | | | | | | |
| No | 23 (76.7%) | 15 (50.0%) | 8 (26.7%) | 9 (52.9%) | 3 (18.8%) | 12 (34.3%) |
| Yes | 7 (23.3%) | 15 (50.0%) | 22 (73.3%) | 8 (47.1%) | 13 (81.2%) | 23 (65.7%) |
| Smoking | | | | | | |
| No | 28 (93.3%) | 19 (63.3%) | 12 (40.0%) | 17 (100.0%) | 8 (50.0%) | 18 (51.4%) |
| Yes | 2 (6.7%) | 11 (36.7%) | 18 (60.0%) | 0 (0%) | 8 (50.0%) | 17 (48.6%) |
| Raw fish eating-habit (Source of *O. viverrini*) | | | | | | |
| No | 24 (80.0%) | 7 (23.3%) | 4 (13.3%) | 13 (76.5%) | 2 (12.5%) | 4 (11.4%) |
| Yes | 6 (20.0%) | 23 (76.7%) | 26 (86.7%) | 3 (17.6%) | 14 (87.5%) | 30 (85.7%) |
| Uncertain | 0 (0.0%) | 0 (0.0%) | 0 (0.0%) | 1 (5.9%) | 0 (0.0%) | 1 (2.9%) |
| History of *O. viverrini* infection | | | | | | |
| No | 30 (100.0%) | 24 (80.0%) | 16 (53.3%) | 16 (94.1%) | 14 (87.5%) | 24 (68.6%) |
| Yes | 0 (0%) | 6 (20.0%) | 11 (36.7%) | 0 (0%) | 2 (12.5%) | 10 (28.6%) |
| Uncertain | 0 (0%) | 0 (0%) | 3 (10.0%) | 1 (5.9%) | 0 (0.0%) | 1 (2.8%) |
| Fermented food eating-habit (Source of nitrosamine) | | | | | | |
| No | 2 (6.7%) | 0 (0%) | 0 (0%) | 0 (0%) | 0 (0%) | 0 (0%) |
| Yes | 28 (93.3%) | 30 (100.0%) | 30 (100.0%) | 17 (100.0%) | 16 (100.0%) | 34 (97.1%) |
| Uncertain | 0 (0%) | 0 (0%) | 0 (0%) | 0 (0%) | 0 (0%) | 1 (2.9%) |
| Cancer stage | | | | | | |
| Stage 1 (%) | 0 (0%) | 0 (0%) | 0 (0%) | 0 (0%) | 0 (0%) | 1 (2.9%) |
| Stage 2 (%) | 0 (0%) | 0 (0%) | 4 (13.3%) | 0 (0%) | 0 (0%) | 0 (0.0%) |
| Stage 3 (%) | 0 (0%) | 0 (0%) | 1 (3.3%) | 0 (0%) | 0 (0%) | 1 (2.9%) |
| Stage 4 (%) | 0 (0%) | 0 (0%) | 25 (83.4%) | 0 (0%) | 0 (0%) | 33 (94.2%) |

normalized by the geometric mean of the positive controls and the geometric mean of the top 100 most highly expressed microRNAs. The cir-miRNA that log2 fold-change (log2FC) >1.5 and $P < 0.05$ were used for differential cir-miRNA expression by using a heat map. The predicted targets of differential cir-miRNA were done by miRSystem (http://miRsystem.cgm.ntu.edu.tw) (*Lu et al., 2012*). For reactome analysis (https://reactome.org/PathwayBrowser/#TOOL=AT) (*Fabregat et al., 2018*), targeted mRNAs were then converted to UniProtKB ID by using gene cards of organism-specific databases (https://www.uniprot.org/uploadlists/) (*The UniProt Consortium, 2023*) and subjected for reactome analysis.

## Validation of miR-423-5p, miR-93-5p, and miR-4532 by RT-qPCR

Plasma circulating RNAs from 200 µl plasma of 68 samples which include healthy volunteer ($N = 17$), OV-infected patiences ($N = 16$), and CCA patients ($N = 35$) were isolated by using miRNeasy Serum/Plasma kit (QIAGEN, Hilden, Germany) according to the manufacturer's instructions except for the following modification: a spike in control (MIRXES, Singapore) was added into the sample lysis buffer and measured by NanoDrop 2000 spectrophotometer and Qubit 4 Fluorometer for total RNA and cir-miRNA, respectively. The 100 ng of isolated RNAs were reversed to cDNAs by using ID3EAL cDNA Synthesis System and ID3EAL RT primers (MIRXES, Singapore). The mixture was incubated at 42 °C for 30 min followed by 95 °C for 5 min. We examined three cir-miRNAs, including miR-423-5p, miR-93-5p, and miR-4532. The cir-miRNA expression was performed by RT-qPCR using ID3EAL miRNA qPCR assay (MIRXES, Singapore) on CFX96 Touch™ Real-time PCR detection system (QIAGEN, Germantown, MD, USA). The cDNA products were conducted according to the manufacturer's protocol. Briefly, 5 µl of 1:10 diluted cDNA was combined with 10 µl ID3EAL miRNA qPCR Master Mix and 2 µl ID3EAL miRNA qPCR assay. The total volume of the mixture product was 20 µl. The incubation of the mixture product was carried out at 95 °C for 10 min, 40 °C for 5 min followed by 40 cycles of 95 °C for 10 s and 60 °C for 30 s. The Cq values were used for analysis. The relative gene expression values for cir-miRNAs were normalized by using spike-in control and compared by using ΔCt. The spike in control is synthetic small RNAs (~22 nt) with sequence distinct from endogenous miRNAs.

## Statistical analysis

The statistical analysis was conducted utilizing IBM SPSS Statistics for Windows, Version 26.0 (IBM Corp., Armonk, NY, USA). For each selected cir-miRNA dataset (miR-423-5p, miR-93-5p, and miR-4532), the mean of $2^{-\Delta Ct}$, along with the corresponding standard deviations (SD), was documented. Prior to conducting a one-way analysis of variance (ANOVA), values identified as extreme outliers by boxplot, defined as those above the third quartile (Q3) plus three times the interquartile range (IQR) or below the first quartile (Q1) minus three times the IQR, were excluded from the dataset. To find differences among the three health status groups, an initial assessment of homogeneity of variance *via* the Levene test was performed. Subsequently, if the obtained *P*-value exceeded 0.05, an ANOVA F-test followed by a Tukey Honestly Significant Difference (HSD) analysis was employed. Conversely, if Levene's test yielded a *P*-value equal to or less than 0.05, indicating that the variances between groups are not equal, an ANOVA Welch test, which is unaffected by unequal variances, was conducted, followed by a Dunnett's C *post hoc* analysis. Graphpad was used for scatter plot and area under curve (AUC) of the receiver operating characteristic (ROC) curve (GraphPad Prism version 10.0.0 for MacOS, GraphPad Software, Boston, Massachusetts, USA, www.graphpad.com). Maximum likelihood ratio was used for selecting cut-off. The diagnostic parameters including sensitivity, specificity, positive predictive value (PPV), negative predictive value (NPV), and accuracy were calculated by MedCalc (https://www.medcalc.org/calc/diagnostic_test.php; Version 22.017; accessed January 13, 2024).

## RESULTS

### Circulating miRNA profiles

At least 428 circulating miRNAs have been identified from plasma using the nCounter® SPRINT Profiler (File S1). Using nSolver Analysis program to analyze the significant up- and down-regulated cir-miRNAs of three differential profiles including OV *vs.* HC, CCA *vs.* HC, and CCA *vs.* OV were listed along with the heat map (Fig. 1). Prediction of CCA *vs.* HC down-regulated cir-miRNA target, the chromosome structure and DNA repair were shown (File S2) while CCA *vs.* HC up-regulated cir-miRNA targeted to cell signaling pathway including Rho GTPase, TGFR family, Interleukin 4, 10, and 13, Autophagy, and DNA repair (File S3). For CCA *vs.* OV up-regulated cir-miRNA targeted to the immune system: Calcineurin activates NFAT and CLEC7A (Dectin-1) induced NFAT activation, and CAM-PDE 1 activation (File S4).

### Investigation of potential circulating miRNA expression by RT-qPCR

To differentiate CCA from HC and OV, the top three cir-miRNAs with the highest fold changes in CCA compared to HC ($P < 0.05$) were selected for further validation as biomarkers for cholangiocarcinoma which include miR-4532 (15.26), miR-93-5p (4.18), and miR-423-5p (3.88) (Fig. 1). Moreover, miR-93-5p was specifically up-regulated in CCA compared to OV (2.19-fold, $P = 0.0492$), which is beneficial for distinguishing CCA from patients infected with OV.

Based on the $2^{-\Delta Ct}$ levels of miR-423-5p, a Welch's ANOVA indicated differences among at least two groups (Welch F [2, 41.495] = 12.469, $P < 0.001$). *Post hoc* multiple comparisons using Dunnett C tests revealed significant differences between the CCA group and both the HC group ($P < 0.05$, 95% CI [0.033499–0.099281]) and the OV group ($P < 0.05$, 95% CI [0.034357–0.100437]). However, no significant difference was observed between the HC group and the OV group ($P > 0.05$, 95% CI [−0.007820 to 0.009833]) (Fig. 2).

The one-way ANOVA indicated differences in the $2^{-\Delta Ct}$ levels of miR-93-5p among at least two groups (F [2, 63] = 3.899, $P = 0.025$). Subsequent Tukey HSD tests revealed a significant difference between the CCA group and the OV group ($P = 0.019$, 95% CI [0.000159–0.002121]). However, no significant differences were observed between the HC group and the OV group ($P = 0.275$, 95% CI [−0.000397–0.001846]) or between the HC group and the CCA group ($P = 0.556$, 95% CI [−0.001377 to 0.000545]) (Fig. 2).

Furthermore, the $2^{-\Delta Ct}$ levels of miR-4532 exhibited a similar trend, indicating differences among at least two groups (Welch F [2, 36.106] = 18.969, $P < 0.001$). The Dunnett C test identified significant differences between the CCA group and both the HC group ($P < 0.05$, 95% CI [0.001294–0.002996]) and the OV group ($P < 0.05$, 95% CI [0.001090–0.002890]). Nevertheless, no significant difference was noted between the HC group and the OV group ($P > 0.05$, 95% CI [−0.000546 to 0.000236]) (Fig. 2).

### Diagnostic potential of circulating miRNA

The AUC of the ROC curve and diagnostic parameters including sensitivity, specificity, PPV, NPV, and accuracy of miR-423-5p, miR-93-5p, and miR-4532 were calculated by

**A. Fold change of cir-miRNAs**

| HC vs CCA (28) | Accession No. | FC | P-value |
|---|---|---|---|
| hsa-miR-4532* | MIMAT0019071 | 15.26 | 0.0128 |
| hsa-miR-93-5p* | MIMAT0000073 | 4.18 | 0.0247 |
| hsa-miR-423-5p* | MIMAT0000093 | 3.88 | 0.0347 |
| hsa-miR-130a-3p | MIMAT0004748 | 3.04 | 0.0310 |
| hsa-miR-21-5p | MIMAT0000425 | 2.29 | 0.0278 |
| hsa-miR-630 | MIMAT0000076 | 1.76 | 0.0295 |
| hsa-miR-320d | MIMAT0003299 | -1.50 | 0.0095 |
| hsa-miR-548ar-3p | MIMAT0018980 | -1.56 | 0.0316 |
| hsa-miR-30e-5p | MIMAT0022266 | -1.56 | 0.0398 |
| hsa-miR-520h | MIMAT0000692 | -1.58 | 0.0012 |
| hsa-miR-30a-5p | MIMAT0002867 | -1.58 | 0.0218 |
| hsa-miR-208b-3p | MIMAT0000087 | -1.58 | 0.0462 |
| hsa-miR-1257 | MIMAT0004960 | -1.59 | 0.0142 |
| hsa-miR-2116-5p | MIMAT0005908 | -1.60 | 0.0011 |
| hsa-miR-378h | MIMAT0011160 | -1.62 | 0.0077 |
| hsa-miR-1253 | MIMAT0018984 | -1.62 | 0.0273 |
| hsa-miR-1234-3p | MIMAT0005904 | -1.66 | 0.0233 |
| hsa-miR-1537-3p | MIMAT0005589 | -1.66 | 0.0301 |
| hsa-miR-648 | MIMAT0007399 | -1.71 | 0.0060 |
| hsa-miR-32-5p | MIMAT0003318 | -1.71 | 0.0178 |
| hsa-miR-1279 | MIMAT0000090 | -1.73 | 0.0032 |
| hsa-miR-3161 | MIMAT0005937 | -1.76 | 0.0445 |
| hsa-miR-656-3p | MIMAT0015035 | -1.78 | 0.0145 |
| hsa-miR-1246 | MIMAT0003332 | -1.80 | 0.0409 |
| hsa-miR-597-5p | MIMAT0002812 | -1.83 | 0.0039 |
| hsa-miR-515-3p | MIMAT0003265 | -1.87 | 0.0413 |
| hsa-miR-613 | MIMAT0002827 | -1.90 | 0.0399 |
| hsa-miR-197-5p | MIMAT0003281 | -2.15 | 0.0044 |

| HC vs OV (7) | Accession No. | FC | P-value |
|---|---|---|---|
| hsa-miR-423-5p* | MIMAT0004748 | 2.19 | 0.0492 |
| hsa-miR-613 | MIMAT0003281 | -1.59 | 0.0030 |
| hsa-miR-450a-1-3p | MIMAT0022700 | -1.60 | 0.0470 |
| hsa-miR-625-5p | MIMAT0003294 | -1.62 | 0.0491 |
| hsa-miR-197-5p | MIMAT0022691 | -1.71 | 0.0367 |
| hsa-miR-1262 | MIMAT0005914 | -2.00 | 0.0109 |
| hsa-let-7f-5p | MIMAT0000067 | -2.16 | 0.0202 |

| OV vs CCA (2) | Accession No. | FC | P-value |
|---|---|---|---|
| hsa-miR-630 | MIMAT0003299 | 1.68 | 0.0114 |
| hsa-miR-215-5p | MIMAT0000272 | 1.52 | 0.0038 |

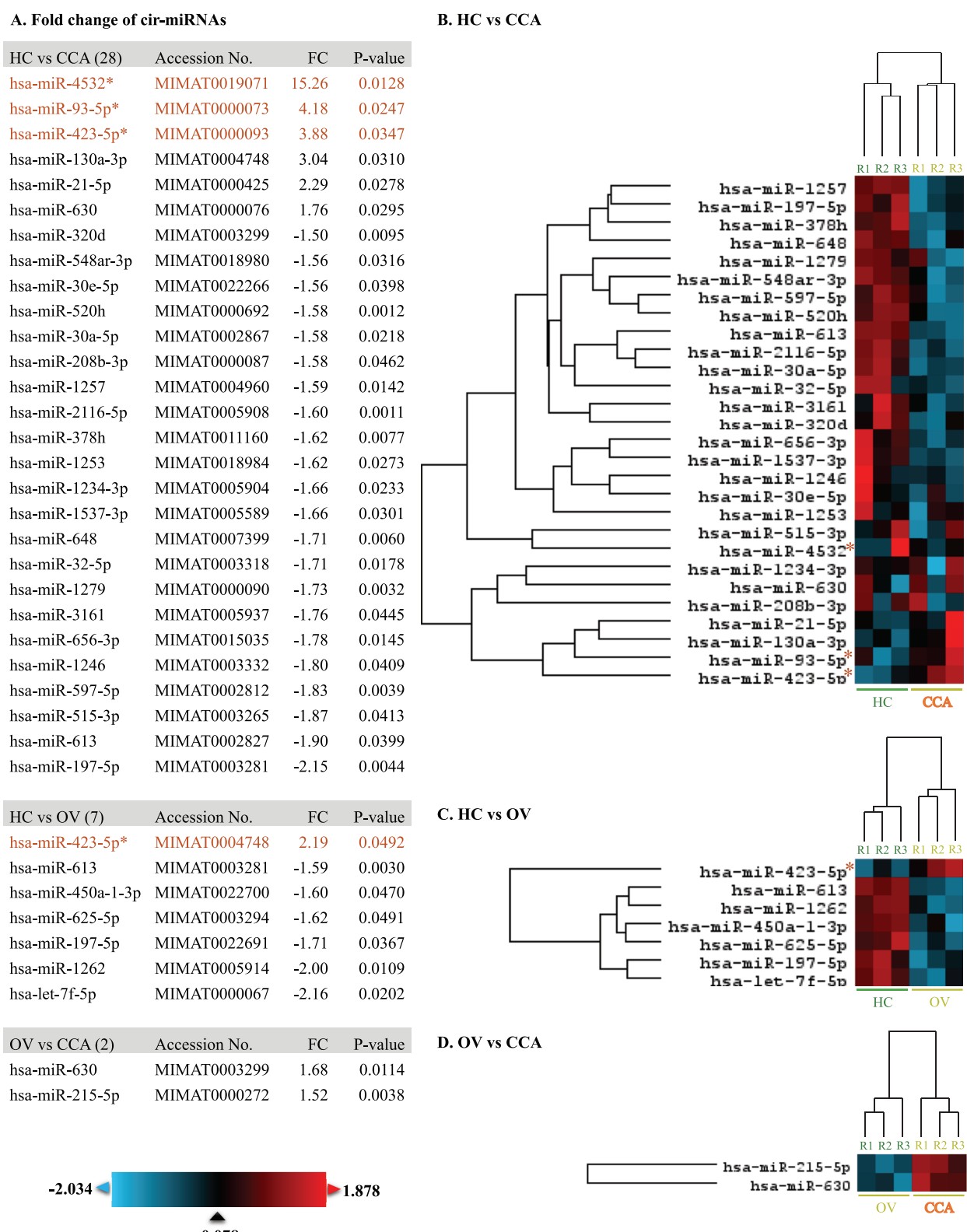

**Figure 1 The heat map of significant up-and down-regulated miRNA profiles.** The expression level was shown from low to high by using blue to red color, respectively. The fold changes of each miRNA were listed. The asterisk indicates the selected cir-miRNAs for validation by RT-qPCR.

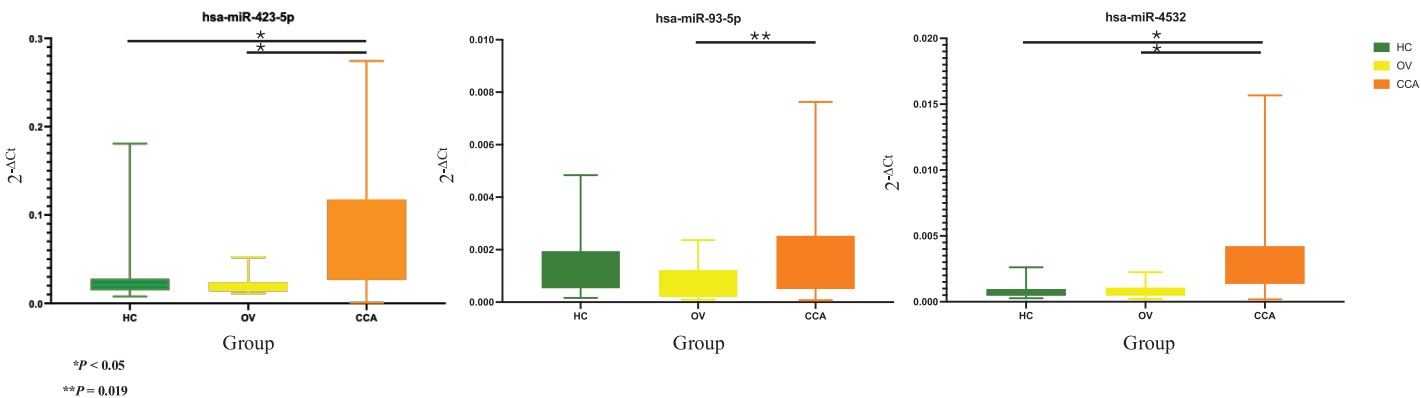

**Figure 2** **Real time PCR.** The box plot of $2^{-\Delta Ct}$ levels of real time RT-qPCR of miR-423-5p (left), miR-93-5p (middle), and miR-4532 (right). The green, yellow, and orange color are used to indicate the HC, OV, and CCA group, respectively. The y-axis showed the $2^{-\Delta Ct}$ levels and groups in the x-axis.                

using Graphpad and MedCalc, respectively, and were summarized in Table 2. The result showed all cir-miRNA could be used for differential CCA from HC and OV groups. The highest AUC of the ROC curve was found for differential CCA from HC group (0.8983) by using miR-4532. The high specificity with low sensitivity were observed in miR-423-5p (cut-off <25.24: 57.14% sensitivity and 96.97% specificity) and miR-93-5p (cut-off <29.32: 20.00% sensitivity and 96.97% specificity). The 77.14% sensitivity and 78.79% specificity were obtained from miR-4532 when using <30.42 as a cut-off. In combination of these three potential cir-miRNAs, the 85.71% sensitivity, 76.92% specificity, 76.92% PPV, 85.71% NPV, and 81.08% accuracy were obtained.

## DISCUSSION

Investigation of cir-miRNA profiles has not only allowed the identification of biomarkers but also facilitated the understanding of possible pathogenesis or carcinogenesis. Prediction of cir-miRNA by reactome targeted to chromosome structure, DNA repair, and transcription regulation, immune system, which was related to previous proteomic profiles wherein checkpoint protein 1 (Chk1) and polymeric immunoglobulin receptor (PIGR) was identified as a potential biomarker (*Phanaksri et al., 2022*; *Prasopdee et al., 2022*). The turning on/off of gene expression will affect the cell cycle, cell apoptosis, cell proliferation, and DNA repair, thus allowing cancer to develop. Therefore, miRNA may plays a vital role in pathogenesis and carcinogenesis and has been utilized as a biomarker in several cancers (*Ahmad et al., 2013*).

Several studies have reported that miRNAs are useful for diagnosis and prognosis approaches for cancers. For cholangiocarcinoma, several miRNAs have been reported as candidates (*Chi et al., 2022*; *Uchihata et al., 2022*; *Høgdall et al., 2022*; *Wada et al., 2022*; *Han et al., 2021*; *Hu et al., 2021*; *Lee et al., 2021*; *Salem et al., 2020*; *Xue et al., 2020*; *Han et al., 2020*). However, the study of liver flukes related CCA was scarce (*Han et al., 2020*; *Plieskatt et al., 2014*, *2015*; *Silakit et al., 2014*). This investigation indicated miR-423-5p, miR-93-5p, and miR-4532 were novel biomarkers for CCA. We suggested patient

**Table 2 Diagnostic parameters.** The diagnostic parameters of miR-423-5p, miR-93-5p, and miR-4532.

| | hsa-miR-423-5p | | | | hsa-miR-93-5p | | | | | hsa-miR-4532 | |
|---|---|---|---|---|---|---|---|---|---|---|---|
| | HC vs. OV | HC vs. CCA | OV vs. CCA | HC&OV vs. CCA | HC vs. OV | HC vs. CCA | OV vs. CCA | HC&OV vs. CCA | HC vs. OV | HC vs. CCA | OV vs. CCA | HC&OV vs. CCA |
| AUC of ROC | 0.5515 | 0.8118 | 0.8482 | 0.8294 | 0.7316 | 0.5983 | 0.7670 | 0.6801 | 0.5294 | 0.8983 | 0.8732 | 0.8861 |
| Cut-off (Ct) | >27.57 | <25.74 | <25.98 | <25.24 | >32.71 | <30.15 | <30.41 | <30.07 | <30.86 | <30.88 | <30.42 | <29.86 |
| Sensitivity | 25.00% | 62.86% | 65.71% | 57.14% | 56.25% | 62.86% | 51.43% | 20.00% | 25.00% | 80.00% | 68.57% | 77.14% |
| Specificity | 94.12% | 93.75% | 93.75% | 96.97% | 94.12% | 88.24% | 93.75% | 96.97% | 94.12% | 94.12% | 93.75% | 78.79% |
| Positive predictive value | 80.00% | 95.65% | 95.83% | 95.24% | 90.00% | 91.67% | 94.74% | 87.50% | 80.00% | 96.55% | 96.00% | 79.41% |
| Negative predictive value | 57.14% | 53.57% | 55.56% | 68.09% | 69.57% | 53.57% | 46.88% | 53.33% | 57.14% | 69.57% | 57.69% | 76.47% |
| Accuracy | 60.61% | 72.55% | 74.51% | 76.47% | 75.76% | 71.15% | 64.71% | 57.35% | 60.61% | 84.62% | 76.47% | 77.94% |

screening with miR-4532 based on its high sensitivity and confirmation of CCA diagnosis by miR-423-5p and miR-93-5p due to specificity.

The up-regulation of miR-93 and miR-423-5p in cholangiocarcinoma tissues was reported (*Wang et al., 2015*; *Zhang et al., 2015*). However, in this study, up-regulation of miR-93 and miR-423-5p was found in plasma samples.

The miR-93-5p has been identified as a potential biomarker for several diseases, such as ocular hypertension (*Tan et al., 2023*), Hepatitis B virus (HBV)-related hepatocellular carcinoma (*Zhou et al., 2022*), polycystic ovary syndrome (*Tan et al., 2022*), breast cancer (*Moghaddam et al., 2022*), oral squamous cell carcinoma (*Aghiorghiesei et al., 2022*), and acute myeloid leukemia (AML) (*Wang et al., 2021b*). The miR-93-5p has been implicated in carcinogenesis across various cancers, including small cell lung cancer (*Liu et al., 2021*), esophageal squamous cell carcinoma (*Su et al., 2022*), and bladder cancer (*Yuan et al., 2023*). In hepatocellular carcinoma (HCC), miR-93-5p was found to be upregulated and promote cell proliferation by downregulating peroxisome proliferator-activated receptor gamma coactivator-1 alpha (PPARGC1A) (*Wang et al., 2021b*). Furthermore, miR-93-5p bound directly to the 3′ untranslated regions of tumor-suppressor genes PTEN and CDKN1A, inhibiting their expression and resulting in enhanced activity of the c-Met/PI3K/Akt pathway. Inhibition of miR-93-5p was found to suppress HepG2 cell proliferation, migration, and colony formation (*Ohta et al., 2014*). In HBV-related HCC, significantly elevated levels of miR-93-5p in patients' plasma and urine suggest its potential as biomarkers for early diagnosis (*Zhou et al., 2022*).

The miR-423 was identified as a biomarker for oral squamous cell carcinoma (*Romani et al., 2021*), heart failure (*Miyamoto et al., 2015*), and diabetes (*Ryu et al., 2020*). The function or target of up-regulated miR-423-5p is not elucidated. However, it likely plays a role in cancer progression, proliferation, and metastasis. miR-423-5p is secreted from the

cell *via* exosomes, transported to the blood circulation, and can fuse with other cells either near or far from the secreting cell. In prostate cancer, miR-423-5p is involved in cancer progression, and enhances migratory and invasive abilities *via* FRMD3, a tumor suppressor gene. In gastric cancer, miR-423-5p is up-regulated and targets TFF1, affecting gastric cancer cell proliferation and invasion (*Liu et al., 2014*). Another possible target of miR-423-5p is p21Cip1/Waf1, which impacts the G1/S checkpoint of the cell cycle and promotes cell growth (*Lin et al., 2011*). miR-423-5p may function as an oncogene and requires further investigation to identify its target in CCA.

Nevertheless, information about miR-4532 is quite limited. MiR-4532 has been reported to be involved in kidney diseases (*Shankar et al., 2023*; *Seo et al., 2023*; *Monteiro et al., 2019*), ovarian cancer (*Hamidi et al., 2021*), leukemia (*Zhao et al., 2019*), and COVID (*Parray et al., 2021*; *Paniri et al., 2021*), but there have been no studies specifically focusing on its role in CCA. The miR-4532 was suggested as a diagnostic biomarker for breast cancer, liposarcoma, and ovarian cancer (*Vora et al., 2024*; *Kohama et al., 2021*; *Hamidi et al., 2021*). In breast cancer, miR-4532 stimulated cell proliferation, anti-apoptosis, disease pathogenesis, disease relapse/metastases and activation of STAT3 and TGFB pathways (*Feng et al., 2018*; *Vora et al., 2024*). In liposarcoma, miR-4532 also enhanced cell proliferation (*Kohama et al., 2021*). In leukemia, miR-4532 repress normal hematopoietic stem cells (HSC) hematopoiesis *via* activation of STAT3 pathway (*Zhao et al., 2019*). Moreover, serum miR-4532 level of type 2 diabetic patients was elevated in a lead time ranging from 0 to 4 years before the diagnosis of liver cancer (*Lee et al., 2021*). miR-4532 in exosomes released by macrophages can be absorbed by vascular endothelial cells, activates the NF-κB P65, that promotes atherosclerosis (*Liu et al., 2022*).

Nevertheless, although the role of these three miRNAs is clearly elucidated in several cancers, their actual role and function should be further investigated, which will be useful for the diagnosis and treatment of CCA. Application of these three cir-miRNAs as potential biomarkers will be very useful for diagnosis of CCA.

## CONCLUSION

The investigation and differential circulating miRNA profile were performed using the nCounter® SPRINT Profiler, through which miR-423-5p, miR-93-5p, and miR-4532 were identified as potential biomarkers for CCA diagnosis. The RT-qPCR was employed to validate these cir-miRNAs, and the results showed that miR-423-5p, miR-93-5p, and miR-4532 were identified as diagnostic biomarkers for differentiating CCA from healthy and *O. viverrini* infected patients.

## ACKNOWLEDGEMENTS

We would like to express our deep gratitude to all participants who enrolled in this study for their devotion and kindness. We would like to thank Dr. Anthicha Kunjantarachot for her guidance and support in the demographic and results analysis.

## Funding

Funding was provided by the Chulabhorn International College of Medicine, Thammasat University, Fund Contract No. T3/2562 to Veerachai Thitapakorn. Additional funding was provided by Thammasat Research Unit in Opisthorchiasis, Cholangiocarcinoma, and Neglected Parasitic Diseases, Thammasat University (TRU-OCN) to Veerachai Thitapakorn and Sattrachai Prasopdee. The funders had no role in study design, data collection and analysis, decision to publish, or preparation of the manuscript.

## Grant Disclosures

The following grant information was disclosed by the authors:
Chulabhorn International College of Medicine, Thammasat University: T3/2562.
Thammasat Research Unit in Opisthorchiasis, Cholangiocarcinoma, and Neglected Parasitic Diseases, Thammasat University (TRU-OCN) to Veerachai Thitapakorn and Sattrachai Prasopdee.

## Competing Interests

Amnat Chukhan is employed by Prima Scientific Co. Ltd. The authors declare that they have no competing interests.

## Author Contributions

- Kittiya Supradit conceived and designed the experiments, performed the experiments, analyzed the data, prepared figures and/or tables, authored or reviewed drafts of the article, and approved the final draft.
- Sattrachai Prasopdee conceived and designed the experiments, performed the experiments, analyzed the data, prepared figures and/or tables, authored or reviewed drafts of the article, and approved the final draft.
- Teva Phanaksri performed the experiments, analyzed the data, prepared figures and/or tables, authored or reviewed drafts of the article, and approved the final draft.
- Sithichoke Tangphatsornruang performed the experiments, analyzed the data, authored or reviewed drafts of the article, and approved the final draft.
- Montinee Pholhelm performed the experiments, analyzed the data, prepared figures and/or tables, authored or reviewed drafts of the article, and approved the final draft.
- Siraphatsorn Yusuk performed the experiments, authored or reviewed drafts of the article, and approved the final draft.
- Kritiya Butthongkomvong conceived and designed the experiments, performed the experiments, authored or reviewed drafts of the article, and approved the final draft.
- Kanokpan Wongprasert conceived and designed the experiments, analyzed the data, authored or reviewed drafts of the article, and approved the final draft.
- Jutharat Kulsantiwong performed the experiments, authored or reviewed drafts of the article, and approved the final draft.

- Amnat Chukan analyzed the data, prepared figures and/or tables, authored or reviewed drafts of the article, and approved the final draft.
- Smarn Tesana conceived and designed the experiments, analyzed the data, authored or reviewed drafts of the article, and approved the final draft.
- Veerachai Thitapakorn conceived and designed the experiments, performed the experiments, analyzed the data, prepared figures and/or tables, authored or reviewed drafts of the article, and approved the final draft.

## Human Ethics

The following information was supplied relating to ethical approvals (*i.e.*, approving body and any reference numbers):

Human ethic committee of Udonthani cancer hospital, Udon Thani, Thailand, granted ethical approval to carry out the study within its facilities (UCH-CT 11/2563).

## Data Availability

The raw data are available in the Supplemental Files.

## Supplemental Information

Supplemental information for this article can be found online at http://dx.doi.org/10.7717/peerj.18367#supplemental-information.

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
