# Peer review of "Differential circulating miRNA profiles identified miR-423-5p, miR-93-5p, and miR-4532 as potential biomarkers for cholangiocarcinoma diagnosis"

_PeerJ, doi:10.7717/peerj.18367_

## Round 0.1 · original submission · Major Revisions

As you can see, all reviewers found your research important. However, they also indicated that the manuscript has several serious issues, among which are statistical analysis, data presentation, and methodological transparency.
Please address concerns of all reviewers and revise manuscript accordingly.

Reviewer 1 ·

Basic reporting

This study presents valuable findings on circulating miRNAs as potential biomarkers for cholangiocarcinoma. The dual-phase design, which involves discovery and validation, is robust, addressing a significant gap in diagnostic aproaches for this cancer. However, the manuscript could benefit greatly from refining its statistical analysis details, enhancing data vizualization, and providing open access to the raw data. Addressing these points would improve the manuscript's clarity and scientific rigor, making the conclusions more relaiable and impactful.

Experimental design

1. More details are needed on the process of plasma pooling in the discovery phase. How was the decision made regarding the volume of plasma to be pooled from each sample? This information is crucial for replicability.
2. The manuscript should include specifics on the spike-in control used during qRT-PCR analysis. Detailing the type and concentration of the spike-in could help in understanding the quantification process better.

Validity of the findings

1. While the manuscript mentions that raw data was supplied, it does not provide a direct link or reference to where this data can be acessed. Ensuring open access to the raw data used in the analyses, as per journal guidelines, is essential for reproducibility and further studies.
2. The rationale for selecting the specific miRNAs for the validation phase is not clearly justified. The manuscript should explain why these miRNAs were choosen out of all the diferentially expressed candidates from the discovery phase.

Additional comments

1. In the description of statistical analysis (line 156-172), it's not clear how the outliers were determined and removed. Please specify the criteria used for identifiying these outliers to ensure the robustness of the statistical conclusions.
2. The manuscript would benifit from an explanation of why Welch’s ANOVA was chosen over a standard ANOVA in certain cases. Detailing the assumptions cheked that led to this decision would strengthen the methodology.
3. The heatmap in Figure 1 would be more informative if it included annotations for the clinical status of each sample (e.g., HC, OV, CCA). This would help in directly vizualizing the correlation between miRNA expression levels and disease conditions.

Reviewer 2 ·

Basic reporting

The manuscript provides some relevant background and context, but the authors do not offer sufficient details on the epidemiology, mechanisms, or relative contributions of liver fluke infections to CCA risk.


The Materials and Methods section should be divided into clear subsections such as Study Cohort, Chemicals and reagents, RNA extraction and quantitative real-time PCR, and Statistical Analysis.

Experimental design

The authors do not clearly state their rationale for focusing on OV infection and CCA. 

The data extraction from medical records, and date range are not stated in the manuscript.

Validity of the findings

The data does not demonstrate the superiority of this approach compared to using the miRNAs individually.

The findings are novel, but replication in larger cohorts would be valuable.

Additional comments

The manuscript lacks essential background information, a clear rationale, and methodological details.

Reviewer 3 ·

Basic reporting

The manuscript by Supradit et al. reports three circulating miRNAs that are differentially expressed in the plasma of cholangiocarcinoma patients. These three miRNAs were initially identified by nCounter® SPRINT Profiler and validated with RT-qPCR in a different plasma cohort. Although the authors provided the raw and processed data that partially support their conclusion, overall, the manuscript lacks depth and clarity.

The introduction needs more details to provide more justification for the study. As stated in the introduction, cfDNA, circRNA, ctDNA, and miRNAs are all important diagnosis markers. The rationale to focus on plasma circulating miRNAs over others should be strengthened. A more comprehensive discussion on the roles of miR-423-5p, miR-93-5p, and miR-4532 is also needed in the discussion section.

In addition, the manuscript should be improved to ensure clarity and avoid inconsistency. For example, in the abstract, plasma circulating miRNA was initially labeled as cir-miRNA but later referred as cf-miRNA.

Experimental design

Overall, the experimental design lacks depth. The manuscript would benefit from a deeper investigation into whether these three circulating miRNAs can also be detected in other published CCA datasets. Understanding the mechanisms behind their upregulation in CCA patients could further enhance the academic value of the paper. Extending the study to include these aspects could provide a more comprehensive understanding of circulating miRNAs in CCA.

Validity of the findings

The heatmap and RT-qPCR results are not sufficient to conclude miR-423-5p, miR-93-5p, and miR-4532 as biomarkers for CCA. In addition, there is inconsistency between the heatmap and RT-qPCR results. For example, miR-423-5p are upregulated in OV samples in discovery cohort but not in validity cohort.

---

## Round 0.2 · accepted · Accept

Reviewers are mostly satisfied by the responses to their critiques. They recommended using higher resolution figure. Hope that this can be done at the proof stage.

Reviewer 1 ·

Basic reporting

After reviewing the revised manuscript, I am pleased with the significant improvements made. The authors have effectively addressed the previous concerns, enhancing the overall quality and ensuring it meets publication standards. I fully support its publication and look forward to its contribution to our field.

Experimental design

NA

Validity of the findings

NA

Reviewer 3 ·

Basic reporting

The authors have addressed my comments. The manuscript would benefit from including higher resolution of figures.

Experimental design

The authors have addressed my comments.

Validity of the findings

The authors have addressed my comments.